**Cite this article:** Browne RK (2025).
Intergenerational justice, the impending sixth
mass extinction, reproduction and advanced
biotechnologies, and humanity's future.
*Cambridge Prisms: Extinction*, **3**, e13, 1–12

anthropogenic impacts; biodiversity
conservation; captive breeding; deextinction;
species extinction

**Corresponding author:**
Robert Kenneth Browne;
Email: robert.browne@gmail.com

# Intergenerational justice, the impending sixth mass extinction, reproduction and advanced biotechnologies, and humanity's future

Robert Kenneth Browne 

Sustainability America, Sarteneja, Belize

## Abstract

Intergenerational justice is the core principle supporting the legacy of benefit toward future generations, including the perpetuation of species and their genetic diversity, as a key component of biospheric sustainability. Thirty percent of Earth's terrestrial habitats are now undergoing protection, biodiversity hotspots are being targeted, and there is increasing community awareness and engagement in conservation. However, the impending sixth mass extinction threatens to drive many species to extinction in the wild, irrespective of these interventions. Earth's biosphere is now undergoing terraforming through ecosystem destruction and modification, urbanization, and agriculture. Therefore, transformative cultural, political, and economic incentives are needed to maximize the legacy of the Earth's biodiversity and biospheric sustainability toward future generations. Reproduction and advanced biotechnologies can perpetuate species and their genetic diversity while also contributing to human and animal health and agricultural production. Advanced reproduction biotechnologies, including genetic engineering and synthetic biology, provide a new horizon for biospheric management, through the de-extinction of ancient species, restoring recently lost species, and maintaining the genetic diversity of extant species through wildlife biobanking. More extensive and inclusive conservation breeding programs and wildlife biobanking resources/facilities are desperately needed to perpetuate more than 3,000 Critically Endangered terrestrial/freshwater species; a goal fully attainable for amphibians and smaller fishes through global inclusion of stakeholders including private caregivers, plausible for freshwater mussels and crayfish, in development for reptiles and birds, and applicable for many mammals. As this capacity develops, many otherwise neglected species that are losing their natural habitat can be perpetuated solely in biobanks, thus enabling the more efficient utilization of resources toward meaningful field conservation primarily through habitat protection. The full potential of reproduction and advanced biotechnologies includes the development of artificial wombs to address the human population crisis and to avoid surrogacy mismatching during species restoration or de-extinction. The use of advanced reproduction biotechnologies for direct human benefit, for species management, and for biospheric sustainability, are subject to evolving ethical and legal frameworks, particularly regarding genetic engineering, transhumanism, and the de-extinction of ancient species.

## Impact statement

This review is unique and impactful through integrating discourses concerning the synergy between intergenerational justice, reproduction and advanced biotechnologies and biobanking, the reduction of species loss from the impending sixth mass extinction, and maintaining biospheric sustainability, human health and reproduction. Intergenerational justice guides the obligations of current generations toward future generations to fulfill our legacy toward biospheric sustainability and species management. The impending sixth mass extinction is conceptualized as resulting from terraforming; the biospheric transformation of Earth to satisfy human desires, which now provides an existential threat to humanity, biospheric sustainability and biodiversity. Reproduction biotechnologies, supported by biobanking, already provide intergenerational justice through assisted reproduction and maintaining genetic diversity in conservation breeding programs. As natural ecosystems are modified beyond recognition advanced reproduction biotechnologies can perpetuate species through their timely restoration of species from wildlife biobanked somatic and germ cells, and through the de-extinction of species from ancient DNA. These developments and utilisation of these biotechnologies contribute to human well-being through enabling biosperic sustainability, and supporting human and animal health, agricultural production, and efficient resource use.



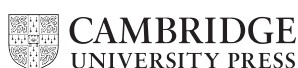

## Introduction

This review is based on intergenerational justice providing the ethical imperative for the provision of biospheric sustainability to maximize utilitarian potential toward humanity's well-being for the foreseeable future (Meyer, 2021). This goal includes species perpetuation facilitated through reproduction and advanced biotechnologies, along with wildlife biobanking, to control reproduction and manage genetic resources for the benefit of wildlife, human health and well-being, agriculture and resource use.

Intergenerational justice recognises the responsibility of current generations to remediate their society's past practices, and address current practices, that could substantially affect future generations; ie. intergenerational justice advances actions and policies to provide future generations access to benefits and the prevention and mitigation of harms [Meyer, 2021]. Intergenerational justice is subject to philosophical and ethical discourses, particularly in respect to previous harms, balancing the rights and preferences of existing people in respect to those of future people, and therefore what the present generation should provide for the future.

Terraforming has traditionally been considered a domain of space colonization. However, this review introduces the concept that Earth's biosphere is now undergoing terraforming through anthropogenic ecosystem destruction and modification, urbanization and agriculture (Khomiak et al., 2024). Consequently, unless emphatic and realistic actions are undertaken, an imminent sixth mass extinction, defined by 75% species/genera extinction over 2 million years beginning with the extinction of a high percentage of Critically Endangered species, and many now Endangered species, is predicted over the next decades and centuries (Barnowsky et al., 2011), irrespective of the protection of their populations, habitats and ecosystems (Fletcher et al., 2023; Willcock et al., 2023; Huggan et al., 2024; Johnson, 2025). Furthermore, many highly biodiversity habitats are unprotected, or already undergoing dramatic defaunation and loss of vegetation biodiversity, with a cascade of species extinctions through global heating alone, even without considering the devastating effects of pollutants, exotic species and pathogens (Bradshaw et al., 2021; IPCC, 2023; Conradi et al., 2024; IUCN, 2025a). No compelling evidence of the initiation of an impending mass extinction was demonstrated by the loss of only 0.45% of genera over the last 500 years (Wiens and Saban, 2025). However, even at the rate of 0.45% per 500 years, 75% of genera would be extinct in 83,500 years, a time period equivalent to only 4% of 2 million years (Wiens and Saban 2025). Higher taxonomy aside, at the species level, the World Wildlife Fund states that we are losing 0.01% to 0.1% species a year. ie. 75% of species 750-7,500 years (WWF, 2025). In either case, these amounts are gross underestimates as biospheric destruction has recently exponentially accelerated. Under these circumstances, the precautionary principle demands that humanity harness all available and potential biotechnologies to assure species perpetuation for the benefit of humanity.

Evidence-based predictions of massive and inevitable ecosystem modifications and species loss (Barnowsky et al., 2011) limits the ability of field-based activities, including habitat protection, rewilding (repopulation), and targeted species conservation to save species and ecosystems (Simkins et al., 2025). The exponentially increasing number of Critically Endangered species is already overwhelming field program capacity to guarantee their survival. However, current biotechnologies along with conservation breeding programs (CBPs) and wildlife biobanking can already provide cost-efficient and reliable management of many otherwise neglected species. The integration of species management, from habitat protection to the use of biotechnologies, demands a ree-valuation of environmental policies and the integration of all available options into an integrated whole (Zeigler, 2023; Browne et al., 2024a, b).

This review coins the term environmental intergenerational justice as the legacy that maximizes benefits for future generations through biospheric sustainability and species conservation/perpetuation in a rapidly changing biosphere (Barry, 1997; Trevis et al., 2018; Spanning, 2021; Pereira et al., 2023; Gergis, 2024). Environmental intergenerational justice also provides an ethical framework to guide conservation investments in species or ecosystem targeting (Treves et al., 2018; Meyer, 2021; Dielenberg et al., 2023). To achieve environmental intergenerational justice, transformative environmental policies including cultural, political, and economic incentives are needed to maximize the legacy of the Earth's biodiversity and biospheric sustainability toward future generations. However, current efforts to transform environmental policy and practice emphasize change within extant political-economic structures, rather than transcending these structures to accommodate a rapidly changing biosphere, multilateral geopolitics, and emerging cultural and ethical norms (Fletcher et al., 2023).

In contrast, the allocation of resources toward reproduction and advanced biotechnologies, along with wildlife biobanks, offers realistic support for environmental intergenerational justice through providing individuals for release during rewilding, repopulation or supplementation programs, through lowering the cost and increasing the reliability of conservation breeding programs (CBPs) and through perpetuating species as their genetic diversity (WAZA 2025; Browne et al., 2024b; Fujihara and Inoue-Murayama, 2024; Chen and Mastromonaco, 2025). Reliable cryopreservation techniques for biobanking are scattered among invertebrates, well established for fish and amphibians, which include half of Critically Endangered species, are under current development for egg-laying reptiles and birds, and well established for many mammals with their naked eggs nurtured by a placenta until birth; artificial insemination with biobanked sperm is used industrially for mammals in aquaculture and for human fertility. Therefore, there are no technical reasons why wildlife biobanking cannot immediately perpetuate the genetic diversity of many species currently in need, and with ongoing development all species in an increasingly threatening and unpredictable future. Besides the current contribution of reproduction technologies to human wellbeing, artificial wombs under development along with genetic engineering, will provide healthy babies to address declining birth rates and reduce the burden and health risks of pregnancy (Gross, 2024; Osmenda, 2024).

Overall, these potentials pose intriguing ethical questions involving what defines humans and transhumans (Battle-Fisher, 2020; Filipova, 2024; Rueda, 2024), the selection and conservation status of species perpetuated solely in biobanks (Cardillo, 2023; Turton-Hughes, 2024; Wienhues et al., 2023; Wilson, 2023; Biasetti et al., 2024), philosophical concepts regarding the transformation from pre-Neolithic pristine nature to Earth's anthropogenic terraforming (Khomiak et al., 2024), and the practicality and consequences of the de-extinction of ancient species, or to a lesser extent the restoration of recently extinct species (IUCN, 2016; Meyer, 2021; Wienhues et al., 2023).

## Historic, cultural, and geopolitical perspectives

From a historical context, this review supports environmental intergenerational justice through remedying species loss caused by our ancient ancestors, to the present, and in the imminent future. The relationship between modern humans, species loss and the environment began with overhunting of megafauna (species over 100 kg in weight) by Neolithic hunter-gatherers, with extinctions peaking in the late Pleistocene ~6,000–20,000 years ago and extends to the present. The pristine megafauna of Australia, the New World, and many islands, became largely extinct during human colonisation; however, much megafauna survived in Africa and Eurasia through coevolution with hominidae species and eventually modern humans (Svenning et al., 2024). However, hunting and human migrations alone do not fully account for ancient megafauna extinctions (Godfrey et al., 2025).

Some megafaunal species still survive in the mid- to low latitudes of Eurasia and North America, for example the auroch (*Bos primigenius*; 1,500 kg) hybridized with domestic livestock, polar bears (*Ursus maritimus*, 800 kg), moose (*Alces* spp., 700 kg), musk ox (*Ovibos moschatus*; 400 kg), wapiti (*Cervus elaphus*; 330 kg), brown bears (*Ursus arctos horribilis*, 300 kg), bison (*Bison*, 920 kg) and Siberian tigers (*Panthera tigris altaica*, 320 kg). No Australian fauna over 90 kg survived Aboriginal occupation, except for the saltwater crocodile (*Crocodylus porosus*, 1,000 kg). Tapir (*Tapirus bairdii*, 250 kg) are the only survivors of the Central and South America megafauna (Svenning et al., 2024; IUCN, 2025b).

The more recent global destruction of biodiversity began as the early city states of Europe and the Middle East, resulted in the first major regional landscape modifications and excessive regional resource use through urbanization, agriculture and forestry, contributing to civilization collapse (Butzer, 2012; Turner and Sabloff, 2012). In contrast, late pre-Columbian agricultural ecosystems of the Maya and Native Indian cultures in the Amazon basin and North America that did not industrialise were sustained for centuries or millenia (Ford, 2024; Glaser et al., 2024). Sustainable agriculture also supported high populations in China, India and Southeast Asia for millennium, with regional megafauna surviving in game parks, relict protected lowland habitats, forested highlands, freshwater wetlands, and mangroves (IUCN, 2025b).

A desperate quest for natural resource exploitation through European colonisation, was regionally driven in Europe by forest destruction, insufficient and degraded agricultural land, and overpopulation (Hughes, 2009; Kaplan et al., 2009). European colonization drove many species to extinction, through resource exploitation, the introduction of exotic species, and targeted extermination. For instance, the transfer of plants and animals from the eighteenth century to the present has transformed the environments globally (Kirchberger, 2020). Colonialism was particularly destructive to biodiversity in the resource-rich bioregions of the New World and Caribbean Islands (Penados et al., 2023), Australia and New Zealand (Dielenberg et al., 2023; Woinarski et al., 2024) through the combination of colonial capitalistic agriculture, mining, indentured workers, slavery, genocide and the planned introduction of exotic species (Kirchberger, 2020). Rapidly increasing human populations in the highly biodiverse regions of Africa, Central and South America, Southeast Asia, and India are now resulting in environmental stress and habitat destruction along with rapid species loss (WWF, 2025; Turvey and Crees, 2019; IUCN, 2025a). From a historic perspective, recent species loss of mammals and birds through hunting and habitat destruction occurred from the mid-1500s, with reptiles from 1800, and then amphibians and fishes from 1900 (Johnson, 2025). Amphibians reached the highest extinction rates of all terrestrial/freshwater vertebrates during the second half of the twentieth century to the present, mainly due to habitat destruction and exotic species including pathogens (Earth.Org, 2025).

## Global heating, ecosystem modification and species loss

Global heating and regional climate shifts threaten the biosphere, species and ecosystems, and provide an existential threat to humanity, as formally recognized as early as the 1970s by world governments (Wang and Mei, 2024). However, global heating is now exponentially increasing due to increasing $CO_2$ and other greenhouse gas emissions, fossil fuel lobbies and compliant governments, failures of renewable energy and carbon sequestration targets, and through reaching of irreversible climate tipping points (Hansen et al., 2023; Browne et al., 2024a b). Furthermore, ecosystems and species, when stressed through unnaturally high temperatures or precipitation, are more susceptible to exotic predators, competitors, pathogens, droughts, fires and floods (Fodham, 2024; Mathes et al., 2025; Zhang et al., 2025). Many highly biodiverse ecosystems are already undergoing irreversible change, and traditional conservation methods are widely failing (Albrich et al., 2020; Turner et al., 2020). For instance, the management of exotic species in the field that threaten whole ecosystems are increasingly expensive, technically difficult, and hard to scale up, especially in remote or low-resource areas (IUCN, 2025c). These threats synergize and are inevitably driving many species to extinction in the wild, even in otherwise pristine habitats, especially species with highly specialized microhabitats, or subject to epidemics by exotic pathogens (Fordham, 2024; Keck et al., 2025).

## Intergenerational justice, biospheric sustainability and species perpetuation

Intergenerational justice supports global initiatives for biospheric sustainability and species perpetuation, as humanity's rights to a supportive environment in the constitutions of 74% of the world's nations (Ladle et al., 2023). These initiatives include international and bioregional strategies for species conservation, habitat protection and wildlife management (Treves et al., 2018; Fletcher et al., 2023; Wilson, 2023; Seet and McLellan, 2024). The inherent value in protecting habitat to provide for species, recreation and other ecosystem services, is widely recognised as a baseline for environmental management (Penjor et al., 2021). However, the ongoing modification of the biosphere and failure of programs to fully address the current biodiversity crisis demands novel and powerful initiatives.

## Habitat protection, rewilding, enclosures and exotic species

Habitat protection, rewilding, exclosures and other field activities provide for species management as formalized in the IUCN One Plan Approach (Ziegler, 2023). The protection of biodiversity hotspots included in 2.5% of Earth's land surface can currently provide habitat for ~45% of bird, mammal, reptile and amphibian species as endemics (Conservation International, 2025), and the planned protection of 30% of terrestrial habitats will provide a haven for many species (COP 16, 2024). Habitat protection, rewilding and exclosures are currently the most reliable and cost-effective techniques for mass species conservation, to provide nature-based

cultural and recreational opportunities, and to support biospheric sustainability (Gammon, 2018; Penjor et al., 2021; COP 16, 2024; Roshier et al., 2020). Greater awareness, concern and conservation incentives, by the public and others in the conservation community, for traditionally less studied wildlife species, 'shadow biodiversity', are encouraged by rewilding projects (Turton-Hughes et al., 2024).

Rewilding is the restoration of habitats damaged through the effects of agriculture, forestry, mining and hydrological changes, such as channelization or wetland destruction (Gammon, 2018). Both habitat and keystone species rewilding can economically and efficiently restore ecosystem function, stabilization and diversification and provide public engagement and recreational opportunities (Gammon, 2018). Multispecies, multiscale habitat suitability models can be used to maximize effectiveness (Penjor et al., 2021).

Ecological restoration is increasing the habitat quality and biodiversity of degraded ecosystems. In contrast, rewilding is often within rural landscapes and builds wild ecosystems as assets and infrastructure for biodiversity conservation, climate adaptation, nature-based enterprises, rural regeneration and more broadly the transition to ecological civilization (Mutillod et al., 2024). Habitat-based rewilding includes reducing or ceasing agriculture and forestry, and strategies like fencing to protect riverine or wetland habitats or remnant vegetation, dechannelizing and removing dams in riverine habitats, and controlling exotic species (Pettorelli et al., 2019; Ramsey et al., 2023). Exclosure of predators through fencing, or through eradication on islands, has proven effective for endangered species (Roshier et al., 2020). Community participation has also proven highly effective in supporting biodiversity conservation and biospheric sustainability through direct engagement in education-based initiatives (Gammon, 2018; Cloyd, 2016).

Projects for rewilded keystone predators include wolves (*Canis lupus*), Californian condors (*Gymnogyps californianus*), Iberian lynx (*Lynx pardinus*), the Tasmanian devil (*Sarcophilus harrisii*) to mainland Australia, where it has been extinct for millennia, and the black-footed ferret (*Mustela nigripes*) restored through cloning, are exciting and popular and can only continue to gain momentum (Cowl et al. 2024). Rewilded keystone browsing species provide habitat engineers including beavers (*Castor fiber*) and bison (*Bison* spp.). Cattle and horses are ecosystem analogues of extinct aurochs and wild horses, and still fill important ecological niches in uncultivated land (Jepson, 2025). Rewilding of deextinct species and the restoration of recently extinct species, offers even more hope for ecosystem stability and biospheric sustainability along with public engagement and support.

### Conservation breeding programs

CBPs in zoos have historically tended toward charismatic mammal and bird species. These initatives have now diversified to include a wide range of threatened mammals, birds, reptiles, amphibians, fishes, and both aquatic and terrestrial invertebrates (Conde et al., 2013). CBPs can also include any self-sustaining captive populations; exemplified by established private caregiver's conservation programs for birds, reptiles and tortoises. Private caregiver's CBPs can address the conservation needs of small invertebrates, amphibians, freshwater fishes, birds, reptiles, and mammals, as they are easily maintained in small areas. Private caregivers have proven willing, capable, and on their own initiative, are maintaining CBPs for an increasing number of species (Browne et al., 2018).

However, misinformed, outdated, or 'official' policies can restrict inclusive participation in species perpetuation. For example, 'model' CBPs for amphibians require a strategy for repopulation into the wild that excludes thousands of species in need and private caregivers (Bradfield et al., 2023; IUCN 2024; Browne et al., 2024a,b). Furthermore, simple reproduction biotechnologies using cryopreserved sperm can, economically and reliably, maintain genetic diversity of most species if sufficient females are available to provide oocytes even from domestic varieties (morphs) (Browne et al., 2024a,b).

### Reproduction and advanced biotechnologies

Conservation can benefit from reproduction and advanced biotechnologies along with biobanks through:

1) Lowering costs and increasing the reliability of CBPs.
2) Fewer individuals needed for each species CBP – more species can be economically housed.
3) Providing genetic engineering for adaptation to:
   a. New environments where global heating strains the physiological limitations of thermoconformers and behavioral thermoregulators, in contrast to physiological thermoregulators.
   b. Pathogens.
4) Perpetuating otherwise neglected species whose habitat is permanently lost through CBPs and biobanks.
5) Managing genetic diversity for species health and survival.
6) Restoring species genetic diversity from private caregiver's collections even from domestic varieties; only one gravid female needed.
7) Restoring species genetic diversity through biobanked sperm and by cloning cryopreserved nuclei.

### Reproduction biotechnologies

The term 'reproduction biotechnologies' refers to biotechnologies that facilitate reproduction and the management of genetic diversity. The most basic reproduction biotechnology is habitat simulation to promote reproduction through temperature and lighting regimes, diet, and enclosure size and substrate. Hormonal stimulation can promote reproduction through injection or implants, through topical skin applications, nasal membrane absorption or oral ingestion. Hormones can also promote spermiation or ovulation for the collection of gametes (Browne et al., 2024a,b). Sperm and oocytes can be sampled for external fertilisation with externally fertilizing amphibians, fishes and invertebrates, or artificial insemination can be used for internally fertilising amphibians (Browne et al., 2019), reptiles (Sandfoss et al., 2024), birds (Assersohn et al., 2021; Almeida and Resende 2023) and mammals (Huijsmans et al., 2023). A mammalian example is where the captive breeding of clouded leopards (*Neofelis nebulosa*) is challenging because of mating incompatibility due to mate fidelity, a high incidence of poor sperm, and unpredictable variability of ovulations in females (Tipkantha et al., 2017). Artificial insemination with either fresh sperm or biobanked cryopreserved sperm has produced an increasing number of genetically diverse clouded leopard progeny (Tipkantha et al., 2017; Nashville Zoo, 2019).

### Advanced reproduction biotechnologies

Advanced reproduction biotechnologies include cloning, cell propagation, and genetic engineering (Shoop et al., 2023; Cowl et al.,

2024), and bioartificial organs including artificial wombs (Shanmugam et al., 2023; Fox-Skelly, 2024). Dependent on the taxon, cloning can be achieved through twinning, embryo splitting, and nuclear transfer. Homocytoplasmic cloning is where the nucleus donor and surrogate female for eggs are from one species. Homocytoplasmic cloning provides wildlife conservation and perpetuation, human fertility, improved livestock and pet restoration. Homocytoplasmic cloning does not produce adults identical to the nucleus donor due to psychological and epigenetic effects (Cowl et al., 2024).

Homocytoplasmic cloning is generally accepted within the traditional conservation community, where cloning and biobanking are increasing the genetic diversity of the northern rhino (*Ceratotherium simum cottoni*) where no living males remain. Eggs from the two remaining females were fertilized using biobanked sperm. The resulting embryos are frozen and biobanked until surrogate mothers from the other subspecies, the southern white rhino (*C. s. simum*), are available. Males could also plausibly be cloned from biobanked tissue (Cowl et al., 2024; Wilder et al., 2024). In 1987, the imminent extinction of the black-footed ferret (*M. nigripes*) promoted a very successful CBP seeding many successful repopulations, resulting in an IUCN status downgrade from Critically Endangered (CR) to Endangered (EN). Black-footed ferrets are vulnerable to pathogens, however, cloning from the biobanked cells of a long-dead female and restoring her unique immune system will increase the species survival and the projects long-term viability (Sullivan, 2024).

Biotechnologies for de-extinction and species restoration, where no males or females remain, can recover species through both homocytoplasmic cloning, and heterocytoplasmic cloning with closely related species, where a nucleocytoplasmic hybrid is created with the donor's nucleus and the surrogate's egg from different species. Heterocytoplasmic cloned frogs have reproduced to the second generation (Browne et al., 2024a). Nuclei-cytoplasmic incompatibility can occur, but cell organelles such as mitochondria can be replaced or genetically engineered to match the donor nucleus (Caicedo et al., 2017). There are second generations in frogs and salamanders from heterocytoplasmic clones but not from other taxa (Browne et al., 2024a).

Genetic engineering through CRISPR enables the precise, efficient, and relatively inexpensive editing of DNA to produce genetically modified organisms to develop new hybrid phenotypes with desirable traits. CRISPR has controversially produced functioning humans from eggs, and CRISPR by injection for disease management and human enhancement at any life stage is under development and application (He, 2020). The first nascent de-extinct species is the ancient dire wolf from the merging of the grey wolf and dire wolf genetics. These first CRISPR puppies are being habituated for wildlife management, are semi-free ranging, and still growing beyond the size of grey wolves to provide a recovered keystone predator (Colossal Biosciences, 2025; Gedman et al., 2025).

The application of these technologies in medicine and wildlife conservation is limited by the need for surrogate mothers, particularly with birds, because fertilisation is deep in the fallopian tubes and where the target cloned species' egg volume is too different to those of the egg donor. Bioartificial wombs enable individuals to be restored, from fresh or cryopreserved eggs or embryos, without the need for a conventional surrogate mother. The use of humidicribs for advanced fetus survival is a medical convention (Fox-Skelly, 2024), and progress on artificial placenta and wombs is predicted to soon nurture early fetus development (Shanmugam et al., 2023; Fox-Skelly, 2024). Artificial placenta could benefit population management in countries with unsustainably low birth rates with China a major developer of robotic artificial placenta for humans, and Colossal for wildlife. Biobanking would also enable the efficient transportation of biomaterials, such as sperm, oocytes, embryos or larvae, on long journeys through space.

## Species restoration, de-extinction, and terraforming

Environmental intergenerational justice includes a moral responsibility, where practicable, to work toward de-extincting lost Pleistocene megafauna and restoring recently lost species. Species restoration is the production of living individuals of a recently extinct species from biobanked living biomaterials. De-extinction is the process of creating nucleocytoplasmic chimeras between extant and extinct species, using DNA and genetic engineering, that are morphological simulacrums, and ecological equivalents of the target species (Odenbaugh, 2023).

Restoration programs for recently extinct species include the Australian thylacine (*Thylacinus cynocephalus*), where the last individuals surviving in zoos were killed by colonial government decree, passenger pigeons (*Ectopistes migratorius*), dodos (*Raphus cucullatus*) and moas (order Dinornithiformes). De-extinction is the production of living individuals of megafauna or iconic fauna from the late Pleistocene. The first late Pleistocene species on the path to full de-extinction is the dire wolf (*Aenocyon dirus*) in 2025, with the first steps toward baby woolly mammoths (*Mammuthus primigenius*) expected in 2027 (Colossal Biosciences, 2025). A range of other extinct fauna may also be de-extincted. As our knowledge of genetics increases, AI-assisted CRISPR promises novel combinations of species genetics to provide display or companion animals, or ecologically functional species in novel anthropogenic environments on Earth or in space. All genetically modified organisms are subject to patent under US laws, and the patenting of de-extinct species using CRISPR, or similar technologies, are apparently included under this patent but this legality has not been tested (Regalado, 2025).

There are a wide range of philosophical, ethical, and legal viewpoints concerning de-extinction and species restoration involve the cultural or ecological roles of de-extinct species in biospheric sustainability (Sandler, 2020; Odenbaugh, 2023). The rewilding of de-extinct species, similarly to rewilding extant species, can support ecosystem restoration, complexity, and integrity.

Unfortunately, misinformation regarding de-extinction presents woolly mammoths and dire wolves as 'ice age' fauna whose habitat is long gone and cannot meaningfully contribute to biospheric sustainability. However, woolly mammoths are prioritized for de-extinction, because their rewilding into northern regions could reduce global heating and permafrost melt, a major threat to biodiversity and biospheric sustainability, through their browsing reducing shrub and tree cover (Revive and Restore, 2025). Dire wolves were generalists and ranged across North and South America in nearly every habitat. Dire wolves are perfect predators to maintain the ecological balance of refaunated northern latitude ecosystems. Dire wolves targeted woolly mammoths, and their predation would keep woolly mammoths wild populations healthy, along with the populations of other remaining northern latitude browsing species (IELC, 2025).

## Biobanking and species perpetuation

Biobanking supports species in the wild and can perpetuate species whose habitat is irretrievably lost (Bolton et al., 2022). The sixth mass extinction has promoted an exponential increase in the number of wildlife biobanking publications since 2010. Biobanking is best conducted regionally to save international transportation costs of biomaterials, to avoid export regulations, and to encourage public support and participation (Chen and Mastromonaco, 2025). Unfortunately, there has been slow progress on the internationalisation of wildlife biobanking (Mukanov et al., 2024). Wildlife biobanking would benefit from: (1) more wildlife biobanks in high biodiversity but low-income countries, (2) focusing publicity on human demographies amenable to biobanking the target species, (3) avoiding research unless specifically targeted, (4) providing excellent biobanking facilities and management and (5) garnering financial support from sponsorship and from species projects (Chen and Mastromonaco, 2025). A globally inclusive biobanking initiative could lobby for wildlife biobanking as a critical global facility deserving funding from major environmental fund managers (Browne et al., 2024a,b; Mukanov et al., 2024).

Cultural and bureaucratic are support is not always evident for the implementation of reproduction and advanced biotechnologies for the benefit of wildlife. International guidelines are not binding and sometimes contradictory, biased, or anachronistic. The IUCN One Plan approach to conserve wildlife supports the use of reproductive biotechnologies including biobanking (Ziegler, 2023). However, the IUCN also favours CBPs only for 'species that can be returned to the wild'. This mandate denies intergenerational justice through undermining support for the biobanking and conservation breeding of, for example, the 880 Critically Endangered amphibian species many of whose habitats will largely become unsupportive over the coming decades (Bradfield et al., 2023). Neglecting these or any species is an unnecessary and tragic loss for future generations (Browne et al., 2024a,b). European Association of Zoos and Aquariums mandates support using reproduction and advanced biotechnologies only to support their current programs, including research (EAZA 2024).

Bias against biotechnologies is evidenced when a 2024 conservation-based review on freshwater mussels' conservation did not mention biobanking the 82 CR species, even though the larvae of other molluscs have been successfully biobanked (Aldridge et al., 2020). Both anachronism and bias are shown in 'Conservation Evidence', in What Works in Conservation, Table 1.13.2, which categorized amphibian sperm biobanking as 'Unlikely to be beneficial' just above the lowest 'Likely to be ineffective or harmful'. Their references ended in 2016 (Smith et al., 2021). In fact, the release of CR amphibians from eggs fertilized with biobanked sperm is already playing an important and increasing role in amphibian species management (Browne et al., 2024a,b). Perhaps the most disingenuous argument against using reproduction and advanced biotechnologies to assure species perpetuation is that their financial or other support will hamper or obstruct field conservation, either by competing for funds or by reducing cultural support. This disingenuous mantra has been repeated over decades by various authors, but has always been unevidenced (EAZA, 2024; Browne et al., 2024a,b). Furthermore, biobanking and deextinction projects are a major supporter of exemplary field projects through expanding community awareness and participation (Browne et al., 2024a,b; Colossal Biosciences, 2025). In the case of amphibians, regional species assessments in Global South countries by local ecologist recommended biobanking for all Critically Endangered species (Browne et al., 2024a,b).

## Biobanking and restoration of Critically Endangered species

Critically Endangered species facing imminent extinction in the wild demand CBPs supported by biobanking to assure their survival (Dee and Spronk, 2025). Freshwater CR molluscs and crustaceans are major taxon in need of reproduction biotechnologies and biobankings. Vertebrates include freshwater fishes, amphibians, terrestrial reptiles, birds, and mammals (Table 1). Efficient wildlife biobanking is based on accurate information concerning the number of regional CR species most in need. This information can then be used to locate biobanking facilities to focus on regional species (Dee and Spronk, 2025).

Institutional support for species CBPs and biobanks also depends on regional emphasis, speciesism and public perception, available facilities and biotechnologies, and importantly, cost balanced against institutional, species, and environmental, benefits and species candidacy. Speciesism is the favouring of one species over another because of virtuous characteristics such as interesting behaviour, colourful or exotic appearance, or rarity, or with these combined. Speciesism in respect to appeal would be expected to be more common in private caregivers' collections than in zoos, as zoos' CBPs would generally favour species dependent on public policy and perception. Speciesism in respect to institutional investment is plausibly best illustrated by over US$20 milllon being invested in a rhinoceros sub-species, the northern white rhinoceros (*Ceratotherium simum cottoni*) in an ongoing project to clone,

**Table 1.** Taxa with substantial publications on the use of reproductive and advanced biotechnologies, biomaterials that survive cryopreservation, the enhancement of genetic diversity through CRISPR in addition to cloning

| Taxa | Cryopreservation | | | | Recovery | |
| --- | --- | --- | --- | --- | --- | --- |
| | Sperm | Oocytes | Somatic cells | Embryos/larvae | CRISPR | Cloning |
| Freshwater mussels | *yes | no | no | *yes | no | no |
| Freshwater crayfish | *yes | no | *yes | *yes | no | no |
| Freshwater fishes | yes | res | yes | no | no | yes |
| Amphibians | yes | no | yes | no | no | yes |
| Birds / Reptiles | yes | res | yes | res | res | res |
| Mammals | yes | yes | yes | yes | yes | yes |

*The possibilities for freshwater mussels and freshwater crayfish are taken from their marine equivalents. Colossal Biosciences res = research by Colossal Biosciences (Colossal Biosciences, 2025).

using the two remaining females from cryopreserved cells, as sperm was never cryopreserved.

This level of funding could have established many CBPs backed by establishing and supporting biobanks in biodiversity regions in the global south countries and plausibly perpetuated scores or hundreds of amphibian and fish species in high benefit projects. Taxonomic speciesism, without evidence, has also been recently published where lower vertebrates, mistakenly categorised as fish and amphibians, rather than correctly being only fish, in comparison to higher vertebrates (correctly being tetrapods, amphibians, reptiles, birds, and mammals), are presented as lacking public appeal (Holt and Comizzoli 2024). Holt and Comizzoli (2024) also consider that spending resources on 'vanishing small' populations, i.e. CE populations, in third-world countries is unrealistic (The term third world refers to countries unaligned with the USA or the Soviet Union during the Cold-War from 1947 to 1991). Holt and Comizzoli (2024) presumably refer to the Global South; a category that includes a spectrum of countries from economic and technical leaders like China and India to economically challenged countries.

The term 'vanishing small populations' presumably refers to Critically Endangered species such as the 825 amphibians or 805 fish, where, for example, more than 400 amphibian species are already included in in-range or out of range CBPs, and where CBPs supported by research and sperm biobanking already exist in Ecuador and Panama and possibly others. In any case, the advancement of CBPs with biobanks will be energised and informed by the participation of enthusiasts ranging from regional private CBPs, to international private caregivers, highly formalised programs in zoos, and their collaborations. The need for a balanced taxonomic approach, impartial to speciesism and geopolitical bias, can only be achieved through a global organisation representing all regional biopolities (Browne et al., 2024a,b).

Data-deficient (DD) species, because of their rarity, have a greater average critical endangerment than other species categories (IUCN, 2025b). However, a low estimate of CR species within DD species, provides a more realistic estimate of the number of CR species than the IUCN assessment. Throughout this review the number of CR species includes an average estimate CR species among DD species. The total number of 2,750 CR terrestrial/freshwater species includes 82 freshwater mussels, 55 freshwater crayfish, 805 freshwater fish, 825 amphibians, 488 reptiles, 224 birds and 271 mammals. More inclusive wildlife conservation breeding programs and wildlife biobanking resources/facilities are needed to perpetuate these species; a goal fully attainable for amphibians and smaller fishes with the inclusion of private caregivers, plausible for mussels and crayfish with ongoing research and development, and applicable for many reptiles, birds, and mammals. However, generalisations based on mammals based on cloning (Holt and Comizzoli, 2024).

Invertebrates for terrestrial/freshwater wildlife biobanking include 82 (17%) CR freshwater mussels and 55 (10%) CR freshwater crayfish (Fluet-Chouinard et al., 2023). No publications describe the use of biobanked material to recover freshwater mussels or freshwater crayfish. Nevertheless, biotechnologies that recover individuals from sperm, embryos, or larvae of commercial marine bivalves could provide for freshwater mussels, and those for marine shrimp larvae for freshwater crayfish are being trialed (Sandeman and Sandeman, 1991; Haagen and Blackburn, 2024; Heres et al., 2022).

Freshwater fishes include 880 CR species and 6% of total species, and amphibians 889 CR species or 11% of total species. These high percentages reflect the destruction of stream and river benthic habitats through siltation, and with amphibians their general dependence on a biphasic aquatic and fully terrestrial life history (Fluet-Chouinard et al. 2023; Luedtke et al., 2023). Most freshwater fishes and amphibians have a common reproduction mode of external fertilization, and the sperm and oocytes of most fishes and amphibians are easily collected, biobanked and individuals produced through external fertilization. Fishes and amphibians' large fatty eggs and early larvae have resisted cryopreservation, and live females are needed to contribute fresh oocytes for fertilization or cloning (Browne et al., 2019, 2024a b). This need is not an impediment and offers the opportunity to involve zoos, other institutions, and privates, to participate by providing oocytes from a few females.

Amniotes, reptiles, birds and mammals, comprise 983 CR species in total. The egg-laying amniotes comprise 712 CR species in total, including 488 (5%) CR reptile species and 224 (2%) CR bird species (number and percentage of total species, respectively). The higher percentage of CE reptile species compared to bird species reflects reptiles' limited mobility and lower thermal homeostasis, particularly in an era of global heating. Both reptiles and birds have a common reproductive mode of internal fertilization and laying amniotic eggs, although some reptiles give birth to live young. Because egg formation begins high in the reproductive tract, the use of reproduction technologies is currently limited to artificial insemination with fresh or cryopreserved sperm (Castillo et al., 2021). This technique is widely used commercially for poultry and experimentally in reptiles, including tortoises (Ravida et al., 2016), snakes (Sandfoss et al., 2024), crocodilians for commercial production (Johnston et al., 2017) and lizards (Campbell et al., 2020). Colossal Biosciences is overcoming the challenges of CRISPR engineering, heterocytoplasmic cloning and development in surrogates, to de-extinct bird species from ancient DNA (Colossal Biosciences, 2025). A technique that would even more easily apply to species restoration from biobanked material.

Mammals include 271 (4%) CR species. Mammals internal development through a placenta make mammals the most amenable group for the cryopreservation of oocytes and embryos and for external fertilization (Bolton et al., 2022). The cryopreservation of mammalian sperm has not yet been achieved for all species (Bolton et al., 2022; Rodger and Clulow, 2022) in external fertilization still being optimised (Oliveira Santos et al., 2022). Mammals are amenable to CRISPR and cloning both for commercial and conservation purposes (Brand et al., 2025). The biobanking of somatic cell lines from tissues is costly and involves animal welfare issues. However, research on establishing pluripotent cell lines from blood is advancing. When established this technique will enable somatic cells of many species to be biobanked economically and efficiently as taking blood is a standard veterinary procedure (Colossal Biosciences, 2025). The longevity of cryopreserved cells for cloning was demonstrated by the production of Endangered Przewalski's Horse, *Equus przewalskii*, from historically cryopreserved cells (Novak et al., 2025).

In summary, sperm can generally be cryopreserved and perpetuate genetic diversity. However, only mammalian oocytes can be cryopreserved. Most species somatic cells can be cryopreserved, and the embryos and larvae of molluscs, crustaceans, and mammals. Genetic engineering through CRISPR has only been widely applied to mammals, but is under research and development for birds and reptiles. Homocytoplasmic cloning has succeeded with freshwater fishes, amphibians, and mammals Table 1. Amphibian and fish sperm are technically the most

suitable for cryopreservation, as simple protocols cover most species irrespective of their reproductive biology with ongoing research continually improving efficacy (Howard et al., 2025; Taheri-Khas et al., 2025). They have also proven amenable to intracytoplasmic sperm injection (ICSI) even when using 20 year old cryopreserved sperm (Péricard et al., 2025). Protocols for birds and reptiles are advanced with artificial insemination achieved in a few species. Some mammals are still challenging despite considerable research expenditure, sometimes over decades. Colossal Bioscience's research potential is predicted to solve many challenges with mammals, reptiles, or birds. These advances, along with current biobanking potentials, especially for freshwater fish and amphibians, cement the foundation of 'species deextinction and restoration' as a new scientific and technical field that challenges the present while providing for the future.

## Ethics, advanced reproduction biotechnologies, and humanities future

The utilization of advanced reproduction technologies raises intriguing ethical issues as generational justice merges toward intergenerational justice. For instance, ethical issues involve the human long-term physical and social consequences of genetic engineering including transhumanism (Rueda, 2024). Other considerations are genetic inequality, social stratification and marginalization, unpredictable consequences of genetic modifications, the commodification of human life, theological concerns with natural processes and the sanctity of life, and a lack of global consensus on regulation to avoid harm (Anomaly and Johnson, 2016).

Some conventional ethical perspectives could be irrelevant to future generations when supplanted by their novel cultural values. For instance, some countries with declining populations are embracing the use robotic bioartificial wombs, or adapting to population decline by using robots to replace humans. Genetic engineering for human enhancement is rapidly gaining traction. Designer babies could have improved health, happiness, and motivation along with high social skills, and fostering these or other virtues could become the ethical norm (Anomaly and Johnson, 2016; Singh et al., 2024). There is a wide range of ethical and legal viewpoints supporting or questioning the role of genetic engineering for the de-extinction of ancient species or the restoration of recently extinct species. The main current ethical issues are the cultural or ecological roles of de-extinct species and animal rights and welfare (Sandler, 2020; Odenbaugh, 2023). In any case, the utilization of advanced reproduction biotechnologies will be driven by future human needs and opportunities rather than by conventional ethical considerations (Anomaly and Johnson, 2016)

A tool to balance ethics, biotechnological research, media and communication is provided by the ETHical ASsessment Tool (Seet and McLellan, 2024).

## Main findings

- Intergenerational justice provides a judicial, ethical, and philosophical foundation for the utilisation of reproduction and advanced biotechnologies.
- The sixth mass extinction will increase exponentially over the coming decades.

- Reproduction technologies and wildlife biobanks are increasingly used for species management.
- There are 82 freshwater mussel species, 55 freshwater crayfishes, 805 freshwater fishes, 825 amphibians, 488 reptiles, 224 birds and 271 mammals in immediate need of biobanking.
- Greater wildlife biobanking resources/facilities are desperately needed for these species.
- Reproduction and advanced biotechnologies provide for species management in wildlife biobanks.
- Advanced reproduction biotechnologies also provide biospheric sustainability through de-extinction or species restoration.
- Reproduction and advanced biotechnologies can also benefit human health, agriculture, and efficient resource utilization.

## Conclusion

Earth's ecosystems are undergoing irreversible anthropogenic changes resulting in terraforming and an impending a sixth mass extinction. However, although some species may be lost, many will survive in the vast network of protected habitats, and those that cannot can be perpetuated through the triad of conservation breeding programs, reproduction and advanced biotechnologies, and biobanks. These species now include invertebrates, freshwater fish and amphibians, and potentially many reptiles, birds, and mammals. Although rarely referred to in the conservation literature, intergenerational justice is a core ethical and legal principle that supports the legacy of benefit toward humanity's future. A legacy that includes the perpetuation of species and their genetic diversity as a key component of biospheric sustainability. Many species can be conserved for the immediate future through habitat protection and rewilding alone. However, reproduction and advanced biotechnologies offer unprecedented opportunities for species management in the wild, for species perpetuation in conservation breeding programs and in biobanks, and for biospheric sustainability. Harnessing the full potentials of these biotechnologies requires a transformative realignment of conservation management based on efficiently and reliably achieving environmental intergenerational justice. Advanced reproduction biotechnologies for wildlife are focused on the de-extinction, restoration of keystone and iconic bird and mammal species, and the development of novel genotypes to provide resistance to environmental threats. A sufficient number of wildlife biobanks must be built and supported in species-diverse and underdeveloped regions. All wildlife biotechnologies contribute to the well-being of humanity through improving health, agricultural production and resource efficacy and, thus, a more sustainable biosphere.

**Open peer review.** To view the open peer review materials for this article, please visit http://doi.org/10.1017/ext.2025.10005.

**Data availability statement.** Data used in this review is publicly available from the IUCN Redlist, or through referenced articles.

**Acknowledgments.** This review acknowledges the countless hours of work by devoted researchers throughout the world who believe in providing for the future of humanity through biospheric sustainability and species perpetuation.

**Author contribution.** This review was written solely by Robert Browne with no external assistance.

**Financial support.** This review has had no financial support.

**Competing interests.** The author is not engaged in research, is semi-retired and has no possible conflict of interest, financial or career interests.

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
