## [Reviewer Report]

I must admit to have struggled a little with this one. As a geneticist with an interest in assisted reproduction and conservation, this paper overlaps with my research interests but is not core to them. As such, I needed, for instance to Google the term “intergenerational justice” (which, to be fair, is fairly self-explanatory). I may therefore be not appreciating the “norm” is a slightly different academic area to myself and this I do acknowledge my potential shortcomings as a reviewer for this paper.

With all this said, I did find it difficult to establish what the purpose of the paper was, and how it adds to knowledge that we already know. I found it making, repeatedly, what I thought to be somewhat obvious statements. I also did not feel that it went either into great scientific depth, nor (and I am less qualified to judge here) any great depth in the field of the social sciences either. The title is quite grandiose and, as such, under this remit could fill several edited volumes. As it stands, it seems to me that the bulk of the paper is concerned with mammalian reproductive technologies, with a nod to other species such as corals, mussels and crayfish.

In the first figure, we learn about extinction rates but this largely focuses only on vertebrates, this would be fine if the authors had set out, from the outset, to focus on vertebrates alone but, as it is, the focus is either much narrower, i.e. on mammals or much broader (corals, mussels, crayfish etc) - I can’t really decide which.

The second figure is a little troubling. I can see the issues these days that no images are free to use and the need to get permission, or pay for the image. As such, the temptation to use AI to draw new images is high. These images do however appear somewhat “cartoon-esque” with the idealised sunny backgrounds. If the authors are going to go down this line, I would recommend re-drawing with a white background and asking a specialist zoologist to confirm the scientific accuracy.

The third figure, I’m sorry but really has to go. These strange idealistic versions of genetic modification, artificial wombs (in space!) and “people sitting around a table” - meant to represent the media really detract form the perceived integrity of the whole article.

Indeed, the article seems to “fall off a cliff” latterly with the discussion of space and cyborgs. These are undoubtedly interesting subjects (I have an interest in space reproduction myself) but they really get away from the core of the article, they dilute it and give the impression of loss of integrity.

As it stands, I think there may be value in this article if it were re-written with the concept of intergenerational justice in mind. A deep dive into the literature pertaining to this topic could then set the scene for how mammalian assisted reproduction strategies (including biobanking, cloning and CRISPR) might be relevant in this context. As such, species such as corals, mussels and crayfish might be mentioned for comparison purposes, but no more.

Similarly, this would then allow space and time to review some of the the success stories of germ cell biobanking such as white rhinos, clouded leopards and black footed ferrets (to name just a few of many examples), all of which need more “air-time” in a paper like this one.

Once fully re-written, a more focussed title, abstract and impact statement might make this paper worthy of publication.

---

## [Reviewer Report]

This manuscript reviews the state of our planet’s sixth mass extinction in the context of past and current conservation efforts, and the wider toolbox of technological approaches including reproductive technologies and their place in supporting efforts to stem the loss of global biodiversity.

The manuscript places the concept of intergenerational justice at the heart of these efforts and adds the biotechnological approaches into this mix, to explore the likelihood of achieving biodiversity targets, and to consider the societal and ethical implications of what these technologies will mean in the near and medium-term future in a rapidly advancing world. The author champions further use of volunteer facilities for biobanking, in order to accomplish the biobanking of a greater number of species before they succumb to extinction.

Overall, this manuscript has value in that it draws together a large number of other referenced work, and in doing so encourages the reader to think about how interconnecting advances create challenges, dilemmas and opportunities.

A few additional points below;

It isn’t entirely clear at the start what the rationale is for searching the IUCN Red List to produce a list of species for biobanking. This needs to be clearer and more effectively integrated into the main introductory sections of the paper.

The points described in lines 465-469 regarding use of artificial wombs and cryopreserved human embryos in space for space colonisation, are, in my view, too off-topic to be sufficiently relevant to the main thrust of the rest of the manuscript that focuses on the sixth mass extinction. Similarly, the section on cyborgs (line 477-491). These sections could be removed and replaced by a clearer bullet-point summary of the manuscript’s main points that are being advocated.

Throughout the manuscript, there are grammatical errors that need attention, mostly involving missing words, see a non-exhaustive list as examples, below;

Examples of grammatical errors:

Line 88: Most megafauna in Africa ‘are’ still extant as ‘they have’ coadapted with the evolution of hominids

Line 115: ‘failures’ of renewable energy

Line 221: Genetic diversity can be perpetuated [???] many males fresh or cryopreserved sperm

Line 270: ‘approaches’

Line 399: ‘others the role of’

Line 451: We live ‘in an’ unstable

---

## [Editor Report]

Thank you for submitting a thought provoking manuscript. The subject is certainly of interest in this era of de-extinction. I have two constructive reviews that both see the benefit of the subject although more work is required to bring the manuscript together. While I do agree with the first reviewer, particularly around tightening up the narrative, I believe these changes can be easily made and therefore have suggested minor correction.

---

## [Editor Report]

Thank you for submitting your revised manuscript/ Myself and the the two previous reviewers have been through your responses to reviewer comments and the corrections made and are happy with the manuscript. I look forward to seeing it published.